

# The Effect of Quadric Shear Zonal Flows and Beta on the Downstream

# Development of Unstable Baroclinic Waves

Yu Ying Yang, Cheng Zhen Guo, Hong Xin Zhang, Jian Song*

*College of Sciences and Inner Mongolia Key Laboratory of Statistical Analysis Theory for Life*

*Data and Neural Network Modeling, Inner Mongolia University of Technology, Hohhot, China*

*Corresponding author address:* Jian Song, College of Sciences and Inner Mongolia Key Labora-

tory of Statistical Analysis Theory for Life Data and Neural Network Modeling, Inner Mongolia

University of Technology, Hohhot, China.

E-mail: songjian@imut.edu.cn





## ABSTRACT

In this paper, the influence of quadric shear basic Zonal flows and $\beta$ on the downstream development of unstable chaotic baroclinic waves is studied from the two-layer model in wide channel controlled by quasi geostrophic potential vorticity equation. Through the obtained Lorentz equation, we focused on the influence of the quadric shear zonal flow (the second derivative of the basic zonal flow is constant) on the downstream development of baroclinic waves. In the absence of zonal shear flow, chaotic behavior along feature points would occur, and the amplitude would change rapidly from one feature to another, that is, it would change very quickly in space. When zonal shear flow is introduced, it will smooth the solution of the equation and reduce the instability, and with the increase of zonal shear flow, the instability in space will increase gradually. So the quadric shear zonal flow has great influence on the stability in space.

Keyword: $\beta$ effect; quadric shear zonal flow; baroclinic instability; Lorentz dynamics



## 1. Introduction

The downstream development of linear and nonlinear instability has a long history in hydrodynamics. In the actual atmosphere, the great development of general large-scale motion is often related to the baroclinic nature of the atmosphere. Therefore, it is necessary to discuss the instability conditions of baroclinic air flow (Stewartson and Stuart, 1971; Hocking et al., 1972). Charney (1947) and Eady (1949) formulated a model baroclinic instability, they indicated that the disturbance viewed in the atmosphere and ocean could be interpreted as a manifestation of baroclinic instability of the basic zonal flows. A simple two-layer model with small vertical scale to remove interference was first introduced by Phillips (1945). Lin (1955) and Drain et al.(1981) studied the stability of undirectional flows when $\beta$ is zero. Drazin et al. (1982) have shown Rossby waves modified by the basic shear in bartropic model. In recent decades, many meteorologists (Pedlosky, 1976; Polvani and Pedlosky, 1988) have made a lot of discussions on it and obtained a broad research topic. In this paper, the influence of zonal shear flow and $\beta$ on the development of the downstream of the slope is studied. Generally, chaotic behavior appears in the unstable baroclinic system, and its performance needs to be studied in the unstable development environment. Although in Lorenz's work (1963), Lorenz equation is used as the truncation model of thermal convection, they can be directly derived in the weak nonlinear baroclinic flow, so for the complete solution of Fourier, no arbitrary truncation is needed, so in the similar problems in the future it can be used at ease. Through the spatial and temporal development of the baroclinic instability waves studied by Pedlosky (2011, 2019), we can see how the sudden spatial variation of the developing disturbance amplitude is caused by the characteristics of Lorentz dynamics. In the chaotic parameter domain, the time change of the system shows that it is extremely unstable to the initial data, so from the perspective of time change alone, the initial data that we evolved for each feature





according to the Lorenz model has slightly different adjacent characteristics.When the adjacent

features of chaos along time and the dynamic development in the downstream coordinate system

are introduced into it, we will get the first-order divergent solution. Because the fast change of the

behavior in the downstream coordinate system is not caused by the range of the system character-

istics developing from parallel to chaos, the impact of chaos is different from the common impact

in the hydrodynamics. Because in the $\beta$ effect, the unstable solution at the origin of the solution

phase plane tends to be shielded from the trajectory, so for the small value of $\beta$, the solution is also

asymptotic to the periodic solution. The $\beta$ parameters are regarded as a smally but importment

disturbance to the dynamic. Without the $\beta$ effect, the two-layer model with uniform vertical shear

is unstable. The stronger the vertical wind shear is, the more favorable it is toproduce the baro-

clinc instability. The basic zonal flow of baroclinic atmosphere with a certain vertical structure

can show the dynamics instability to the disturbance. Section 2 of the paper derives the governing

equations. Section 3 of the paper gives an example of hypothetical behavior. In the concluding

section, section 4, the implication of the results is discussed.

## 2. Formulation

The standard, two-layer, quasi-geostrophic potential vorticity nondimensional equations (Ped-

losky, 1987; Matthew Spydell et al, 2002; Vallis, 2006)

$$\frac{\partial}{\partial t}[\nabla^2 \psi_n + F(-1)^n(\psi_1 - \psi_2)] + J[\psi_n, \nabla^2 \psi_n + F(-1)^n(\psi_1 - \psi_2) + \beta y] = -r\nabla^2 \psi_n, \quad (2.1)$$

where $n = 1, 2$, the rotational Froude number can be expressed as $F = f^2 L^2/g'D$, $f$ is the Cori-

olis parameter, $L$ represents a characteristic length and $g'$ is the reduced gravity, $D$ is the equal

depth of layers. $\beta = \frac{df}{dy}$ is a constant. $r = (\nu f/2)^{1/2}L/(UD)$ represents dissipation parameter.

Velocities have been by a characteristic velocity $U$ of the initial basic flow, $\nu$ is the kinematic





viscosity. $J(a,b) = a_x b_y - a_y b_x$ is the nondimensional Jacobian operator, where subscripts denote

differentiation. The coordinate $x$ is in the downstream direction while $y$ measures distance across

the stream.

In order to facilitate, use the barotropic stream functions $\psi_B = \frac{1}{2}(\psi_1 + \psi_2)$ and baroclinic stream

functions $\psi_T = \psi_1 - \psi_2$ to describe the equations. In the problem to be considered, the basic state

is composed of the quadric shear basic zonal flows with a barotropic and baroclinic component in

each layer, the streamfunctions are

$$\psi_B = -\int_0^y U_B(y')dy' + \varphi_B(x,y,t), \tag{2.2a}$$

$$\psi_T = -\int_0^y U_T(y')dy' + \varphi_T(x,y,t). \tag{2.2b}$$

Where $U_B$ and $U_T$ are related to latitude $y$ and the functions $\varphi_B$ and $\varphi_T$ are the barotropic and

baroclinic perturbation streamfunctions. From equations (2.1), the perturbations $\varphi_B, \varphi_T$ satisfy

$$(\frac{\partial}{\partial t} + U_B \frac{\partial}{\partial x})\nabla^2 \varphi_B + \frac{U_T}{4}\frac{\partial}{\partial x}\nabla^2 \varphi_T + (\beta - \frac{d^2 U_B}{dy^2} - \frac{1}{2}\frac{d^2 U_T}{dy^2})\frac{\partial \varphi_B}{\partial x} + J(\varphi_B, \nabla^2 \varphi_B) + \frac{1}{4}J(\varphi_T, \nabla^2 \varphi_T)$$

$$= -r\nabla^2 \varphi_B, \tag{2.3a}$$

$$(\frac{\partial}{\partial t} + U_B \frac{\partial}{\partial x})(\nabla^2 \varphi_T - 2F\varphi_T) + U_T \frac{\partial}{\partial x}(\nabla^2 \varphi_B + 2F\varphi_B) + (\beta - \frac{d^2 U_B}{dy^2} - \frac{1}{2}\frac{d^2 U_T}{dy^2})\frac{\partial \varphi_B}{\partial x}$$

$$+ J(\varphi_T, \nabla^2 \varphi_B) + J(\varphi_B, \nabla^2 \varphi_T - 2F\varphi_T) = -r\nabla^2 \varphi_T. \tag{2.3b}$$

where since the upper and lower basic zonal flow are quadric shear, $\frac{d^2 U_B}{dy^2}$ and $\frac{d^2 U_T}{dy^2}$ are constants.

$F$ and $F_c$ are the same as employed in Pedlosky(2019), give the critical curve of instability in the

form of lowest order as a relation between $F_c$, the critical value of $F$ , that is,

$$F_c = \frac{K^2}{2} + \frac{rK^2/k}{2U_T}. \tag{2.4}$$

where the wave number $K^2 = k^2 + l^2$.

For small values of $r$ the minimum occurs at very long wavelengths and we need to consider the

scale of the problems variables. The following assumptions:





(i) The basic flow is only slightly super-critical with respect to $F$ so that

$$F = F_c + \Delta, \Delta \leq 1,$$

(ii) The absolute potential vorticity gradient of the layer model and dissipation are also small,

$$\beta - \frac{d^2 U_B}{dy^2} - \frac{1}{2}\frac{d^2 U_T}{dy^2} = O(\Delta^{\frac{1}{2}}),$$

(Samuel. F. Potter et al, 2013; Mathew T. Gliatto et al, 2019),

$$r = O(\Delta).$$

If $U_B, U_T$ are constants, $\beta = O(\Delta^{\frac{1}{2}})$ (Pedlosky,2019).

(iii) The processes of the generated disturbance systems, such as the slowly varying trough systems and cyclones after being generated in the real atmosphere and ocean, are carried on more slowing than their generating processes, therefore the solution of the equations (2.3a,b) will be a function of "fast" and "slow" space and time variables. In such case, using $\xi$ to represent a new fast spatial coordinate, $X$ to represent a new slow space coordinate, $\tau$ to represent a new fast time coordinate and $T$ to represent a slow time coordinate, each defined by

$$\xi = \Delta^{\frac{1}{2}}x, X = \Delta x, \tag{2.5a}$$

$$\tau = \Delta^{\frac{1}{2}}t, T = \Delta t, \tag{2.5b}$$

We have

$$\frac{\partial}{\partial x} = \Delta^{\frac{1}{2}}\frac{\partial}{\partial \xi} + \Delta\frac{\partial}{\partial X}, \tag{2.6a}$$

$$\frac{\partial}{\partial t} = \Delta^{\frac{1}{2}}\frac{\partial}{\partial \tau} + \Delta\frac{\partial}{\partial T}. \tag{2.6b}$$


<sub>93</sub> The perturbations streamfunctions $\varphi_B, \varphi_T$ will expand the progressive series in the small ampli-

<sub>94</sub> tude, $\varepsilon = O(\Delta^{\frac{1}{2}})$ of the perturbation(Pedlosky, 2019, Vallis, 2006)

$$\varphi_B = \varepsilon(\varphi_B^{(0)} + \varepsilon\varphi_B^{(1)} + \varepsilon^2\varphi_B^{(2)} + ...), \tag{2.7a}$$

$$\varphi_T = \varepsilon(\varphi_T^{(0)} + \varepsilon\varphi_T^{(1)} + \varepsilon^2\varphi_T^{(2)} + ...). \tag{2.7b}$$

<sub>95</sub> Substituting(2.7a,b) into (2.3a,b), we obtain at leading order. At the lowest order in $O(\varepsilon)$ obtaining

<sub>96</sub> the results with a linear relationship,

$$\varphi_B^{(0)} = A(X,T)e^{ik(\xi-\tau c)}\sin\pi y + *,$$

$$\varphi_T^{(0)} = 0, c = U_B + \frac{1}{\pi^2}(\frac{d^2U_B}{dy^2} + \frac{1}{2}\frac{d^2U_T}{dy^2}), F_c = \frac{l^2}{2}, l = \pi. \tag{2.8a-e}$$

<sub>97</sub> where * denotes the complex conjugate of the preceding expression.

<sub>98</sub> At the next order in $O(\varepsilon^2)$ we get an expression for the baroclinic perturbation,

$$\varphi_T^{(1)} = \frac{4}{kU_T}[i(\frac{\partial}{\partial T} + U_B\frac{\partial}{\partial x})A + \frac{ir}{\Delta}A + \frac{k}{\Delta^{\frac{1}{2}}\pi^2}(\beta - \frac{d^2U_B}{dy^2} - \frac{1}{2}\frac{d^2U_T}{dy^2})A]$$

$$\times e^{ik(\xi-c\tau)}\sin\pi y + * + \Phi(X,y,T), \tag{2.9}$$

<sub>99</sub> In (2.9), the final term $\Phi(X,y,T)$ is the baroclinic correction to the mean flow and is a function of

<sub>100</sub> only the slow space-time variables $X$ and $T$, as well as $y$.

<sub>101</sub> According to the above expressions, the nonlinear interaction terms, namely the Jacobian of the

<sub>102</sub> next order, can be calculated and obtain as the governing equation for $\Phi$.

$$(\frac{\partial}{\partial T} + U_B\frac{\partial}{\partial x})(\frac{\partial^2\Phi}{\partial y^2} - 2F_c)\Phi + \frac{r}{\Delta}\frac{\partial^2\Phi}{\partial y^2} = \frac{\varepsilon}{\Delta^{\frac{1}{2}}}\frac{4\pi^3}{U_T}(\frac{\partial}{\partial T} + U_B\frac{\partial}{\partial x} + \frac{2r}{\Delta})|A|^2\sin2\pi y. \tag{2.10}$$

<sub>103</sub> As Pedlosky(2013,2019) gives, as long as $\varepsilon \leq \Delta$ is a basic presumption, which in turn implies

<sub>104</sub> that a solution to (2.10) proportional to $\sin2\pi y$ , is appropriate. Hence a solution of the form

<sub>105</sub> $\Phi = P(X,T)\sin2\pi y$ (Pedlosky, 2011,2019) leads to the governing equation for $P(X,T)$,

$$(\frac{\partial}{\partial T} + U_B\frac{\partial}{\partial x})P + \frac{4r}{5\Delta}P = -\frac{\varepsilon}{\Delta^{\frac{1}{2}}}\frac{4\pi}{5U_T}(\frac{\partial}{\partial T} + U_B\frac{\partial}{\partial x} + \frac{2r}{\Delta})|A|^2. \tag{2.11}$$





After the equation is modified by the baroclinic mean flow, the solvable condition of $O(\Delta^{3/2})$
can be determined by the evolution governing equation of amplitude $A$. After we obtain

$$\left(\frac{\partial}{\partial T} + U_B \frac{\partial}{\partial x}\right)^2 A + \frac{3}{2}\left(\frac{r}{\Delta} - i\frac{k(\beta - \frac{d^2 U_B}{dy^2} - \frac{1}{2}\frac{d^2 U_T}{dy^2})}{\Delta^{\frac{1}{2}}\pi^2}\right)\left(\frac{\partial}{\partial T} + U_B \frac{\partial}{\partial x}\right)A - \sigma^2 A - \frac{\varepsilon}{\Delta^{\frac{1}{2}}}\frac{k^2 U_T \pi}{3}AP = 0,$$

(2.12)

where

$$\sigma^2 = \overline{\sigma}^2 - \frac{ir}{\Delta}\frac{k(\frac{d^2 U_B}{dy^2} + \frac{1}{2}\frac{d^2 U_T}{dy^2})}{\pi^2 \Delta^{\frac{1}{2}}} - \frac{k^2(\frac{d^2 U_B}{dy^2} + \frac{1}{2}\frac{d^2 U_T}{dy^2})^2}{2\pi^4 \Delta},$$

$$\overline{\sigma}^2 = \frac{(2-k^2)k^2 U_T^2}{8\pi^2} - \frac{r^2}{2\Delta^2} + \frac{ir}{\Delta}\frac{k\beta}{\pi^2\Delta^{\frac{1}{2}}} + \frac{k^2\beta}{2\pi^4\Delta}.$$

Let

$$T' = \sigma T, X' = \frac{\sigma X}{U_B}, A = A_0 A', P = P_0 P', b = \overline{b} - \frac{k(\frac{d^2 U_B}{dy^2} + \frac{1}{2}\frac{d^2 U_T}{dy^2})}{\sigma \Delta^{\frac{1}{2}}\pi^2},$$

where (Pedlosky, 2019)

$$P_0 = \frac{3\sigma^2 \Delta^{1/2}}{\varepsilon k^2 U_T \pi}, A_0^2 = \frac{15\sigma^2 \Delta}{4k^2 \varepsilon^2 \pi^2}, \gamma = \frac{r}{\Delta}\sigma, \overline{b} = -\frac{k\beta}{\sigma \Delta^{\frac{1}{2}}\pi^2}$$

the governing equations (2.11) and (2.12) to be rewritten (after dropping primes from the new
dependent variables) as

$$\left(\frac{\partial}{\partial T} + \frac{\partial}{\partial X}\right)^2 A + \frac{3}{2}(\gamma + ib)\left(\frac{\partial}{\partial T} + \frac{\partial}{\partial X}\right)A - A(1+P) = 0,$$

(2.13a)

$$\left(\frac{\partial}{\partial T} + \frac{\partial}{\partial X}\right)P + \frac{4}{5}\gamma P = -\left(\frac{\partial}{\partial T} + \frac{\partial}{\partial X} + 2\gamma\right)|A|^2.$$

(2.13b)

We let $P = -|A|^2 - R$, equations (2.13a,b) yielding

$$\left(\frac{\partial}{\partial T} + \frac{\partial}{\partial X}\right)^2 A + \frac{3}{2}(\gamma + ib)\left(\frac{\partial}{\partial T} + \frac{\partial}{\partial X}\right)A - A + A(|A|^2 + R) = 0,$$

(2.14a)

$$\left(\frac{\partial}{\partial T} + \frac{\partial}{\partial X}\right)R + \frac{4}{5}\gamma R = \frac{6}{5}\gamma |A|^2,$$

(2.14b)

as our final evolution equations. The amplitude $A$ is complex, with real and imaginary parts, so let

$$A(X,T) = A_r(X,T) + iA_i(X,T),$$

(2.15)

Substitution of equation (2.15) into equation (2.14a) lead to

$$(\frac{\partial}{\partial T} + \frac{\partial}{\partial X})^2 A_r + \frac{3}{2}(\frac{\partial}{\partial T} + \frac{\partial}{\partial X})(\gamma A_r - bA_i) - A_r + A_r(|A|^2 + R) = 0, \tag{2.16a}$$

$$(\frac{\partial}{\partial T} + \frac{\partial}{\partial X})^2 A_i + \frac{3}{2}(\frac{\partial}{\partial T} + \frac{\partial}{\partial X})(\gamma A_i + bA_r) - A_i + A_i(|A|^2 + R) = 0. \tag{2.16b}$$

We finally obtain five equations,

$$(\frac{\partial}{\partial T} + \frac{\partial}{\partial X})A_r = \bar{A}_r,$$

$$(\frac{\partial}{\partial T} + \frac{\partial}{\partial X})A_i = \bar{A}_i,$$

$$(\frac{\partial}{\partial T} + \frac{\partial}{\partial X})\bar{A}_r + \frac{3}{2}\gamma\bar{A}_r - \frac{3}{2}b\bar{A}_i - A_r + A_r(|A|^2 + R) = 0,$$

$$(\frac{\partial}{\partial T} + \frac{\partial}{\partial X})\bar{A}_i + \frac{3}{2}\gamma\bar{A}_i + \frac{3}{2}b\bar{A}_r - A_i + A_i(|A|^2 + R) = 0,$$

$$(\frac{\partial}{\partial T} + \frac{\partial}{\partial X})R + \frac{4}{5}\gamma R = \frac{6}{5}\gamma|A|^2. \tag{2.17a-e}$$

Defining the characteristic coordinate $s$ by the differential relations(Pedlosky, 2011, 2019)

$$\frac{\partial}{\partial T} + \frac{\partial}{\partial X} = \frac{d}{ds}, \tag{2.18}$$

(2.17a-e) can be written as the set of first order ordinary differential equations

$$\frac{dA_r}{ds} = \bar{A}_r,$$

$$\frac{dA_i}{ds} = \bar{A}_i,$$

$$\frac{d\bar{A}_r}{ds} + \frac{3}{2}\gamma\bar{A}_r - \frac{3}{2}b\bar{A}_i - A_r + A_r(|A|^2 + R) = 0,$$

$$\frac{d\bar{A}_i}{ds} + \frac{3}{2}\gamma\bar{A}_i + \frac{3}{2}b\bar{A}_r - A_i + A_i(|A|^2 + R) = 0,$$

$$\frac{dR}{ds} + \frac{4}{5}\gamma R = \frac{6}{5}\gamma|A|^2. \tag{2.19a-e}$$

This set of ordinary differential equations with zonal shear flow on the $\beta$-plane, are of the form of

the well known Lorenz equations.




## 3. Results

Since equation (2.14) is affected by the boundary condition $X = 0$, we choose as

$$A(0, T = T_0) = a \sin 2\pi T / T_{period} \tag{3.1}$$

Where $T_{period}$, $a$, $\gamma$, $b$ will be given (Pedlosky, 2019).

When $\gamma$ is sufficiently small,the Lorenz dynamics along the characteristics of the partial differential equations of (2.14) produced chaotic solutions. For development problems in space and time , resulting in a value of $A$ at a given time, which changes suddenly with $X$.

In Fig.1. When $b = 0.4$, the instability of the real part of $A$ is relatively strong. When $b$ increases to 1.2, the instability of the real part of $A$ gradually decreases.When $b = 6$, it can be seen that when $b$ is large enough, the real part of $A$ tends to be stable, indicating that zonal shear flow enhances the stability of the real part of $A$.

In Fig.2. When $b$ is small, the real part of $R$ tends to be stable, and when $R$ suddenly increases to 6, the instability of the real part of $R$ increases, indicating that the zonal shear flow causes the instability of the real part of $R$. When the second derivative of zonal flow is introduced into the equation, it can be found that, with the change of time, zonal shear flow reduces the instability of the real part $A$ and enhances the instability of the real part $R$ to ensure the balance of the system.

## 4. Discussion

The chaotic behavior of weakly nonlinear and slightly unstable baroclinic instability is strongly influenced by the zonal shear flow and planetary $\beta$ effect. When we introduce zonal shear flow it reduces this instability. As can be seen from our diagram, the solution is very smooth for a short period of time, but as time goes on and features lengthen, chaos begins to emerge with its own features, forcing it to approach a constant after a period of time.The condition of a smooth





change at the origin will, after a fixed time, at a certain distance from the origin, the amplitude

will change rapidly from one feature to another, that is, it will change very rapidly in space. Due

to the chaotic behavior along the characteristic lines in the downstream coordinate system and

in the slow coordinate system, the solutions of the adjacent characteristic lines, although very

close, still diverged in the first order, which led to the abrupt change of the spatial variables of the

system.Introducing the second derivative of zonal shear flow can eliminate chaos and smooth the

solution in space.

Although the suddenness of the solution behavior of (2.14) in space is meaningful for all systems

controlled by the Lorentz equations, for our weakly nonlinear system, it means the separation

between the expected slow behavior in space and the slow behavior in time. Therefore, we need

to carry out further in-depth research, in the future work to promote, research.

*Acknowledgments.*    This study were supported by Projects 11362012, 11562014 and 41465002 of

the National Natural Science Foundation of China, Project of 2018LH04005 the Natural Science

Foundation of Inner Mongolia.

APPENDIX

**Detailed derivation of the perturbation streamfunctions $\varphi_B$ and $\varphi_T$ equations**

This appendix we derive the equation (2.3) in detail. The barotropic and baroclinic steamfunctions

$$\psi_B = \frac{1}{2}(\psi_1 + \psi_2), \tag{A.1a}$$

$$\psi_B = \frac{1}{2}(\psi_1 + \psi_2), \tag{A.1b}$$





where

$$\psi_B = -\int_0^y U_B(y')dy' + \varphi_B(x,y,t), \tag{A.2a}$$

$$\psi_T = -\int_0^y U_T(y')dy' + \varphi_T(x,y,t). \tag{A.2b}$$

When $n = 1, 2$ Eq.(2.1)

$$\frac{\partial}{\partial t}[\nabla^2\psi_1 - F(\psi_1 - \psi_2)] + J[\psi_1, \nabla^2\psi_1 - F(\psi_1 - \psi_2) + \beta y] = -r\nabla^2\psi_1, \tag{A.3a}$$

$$\frac{\partial}{\partial t}[\nabla^2\psi_2 + F(\psi_1 - \psi_2)] + J[\psi_2, \nabla^2\psi_2 + F(\psi_1 - \psi_2) + \beta y] = -r\nabla^2\psi_2. \tag{A.3b}$$

We insert Eqs.(A.1) into Eq.(A.3) to obtain the perturbation streamfunctions $\varphi_B, \varphi_T$, respectively,

$$\frac{\partial}{\partial t}[\nabla^2\varphi_B + \frac{1}{2}\nabla^2\varphi_T - F\varphi_T] + (U_B + \frac{1}{2}U_T)\frac{\partial}{\partial x}\nabla^2\varphi_B + (\frac{1}{2}U_B + \frac{1}{4}U_T)\frac{\partial}{\partial x}\nabla^2\varphi_T$$
$$-(\frac{d^2U_B}{dy^2} + \frac{1}{2}\frac{d^2U_T}{dy^2})\frac{\partial\varphi_B}{\partial x} - (\frac{1}{2}\frac{d^2U_B}{dy^2} + \frac{1}{4}\frac{d^2U_T}{dy^2})\frac{\partial\varphi_T}{\partial x} + J(\varphi_B, \nabla^2\varphi_B) + \frac{1}{4}J(\varphi_T, \nabla^2\varphi_T)$$
$$+\frac{1}{2}J(\varphi_B, \nabla^2\varphi_T) + \frac{1}{2}J(\varphi_T, \nabla^2\varphi_B) - FJ(\varphi_B, \varphi_T) - FU_B\frac{\partial\varphi_T}{\partial x} + FU_T\frac{\partial\varphi_B}{\partial x} + \beta(\frac{\partial\varphi_B}{\partial x} + \frac{1}{2}\frac{\partial\varphi_T}{\partial x})$$
$$= -r(\nabla^2\varphi_B + \frac{1}{2}\nabla^2\varphi_T - \frac{dU_B}{dy} - \frac{1}{2}\frac{dU_T}{dy}), \tag{A.4a}$$

$$\frac{\partial}{\partial t}[\nabla^2\varphi_B - \frac{1}{2}\nabla^2\varphi_T + F\varphi_T] + (U_B - \frac{1}{2}U_T)\frac{\partial}{\partial x}\nabla^2\varphi_B - (\frac{1}{2}U_B - \frac{1}{4}U_T)\frac{\partial}{\partial x}\nabla^2\varphi_T$$
$$-(\frac{d^2U_B}{dy^2} + \frac{1}{2}\frac{d^2U_T}{dy^2})\frac{\partial\varphi_B}{\partial x} + (\frac{1}{2}\frac{d^2U_B}{dy^2} + \frac{1}{4}\frac{d^2U_T}{dy^2})\frac{\partial\varphi_T}{\partial x} + J(\varphi_B, \nabla^2\varphi_B) + \frac{1}{4}J(\varphi_T, \nabla^2\varphi_T)$$
$$-\frac{1}{2}J(\varphi_B, \nabla^2\varphi_T) - \frac{1}{2}J(\varphi_T, \nabla^2\varphi_B) + FJ(\varphi_B, \varphi_T) + FU_B\frac{\partial\varphi_T}{\partial x} - FU_T\frac{\partial\varphi_B}{\partial x} + \beta(\frac{\partial\varphi_B}{\partial x} - \frac{1}{2}\frac{\partial\varphi_T}{\partial x})$$
$$= -r(\nabla^2\varphi_B - \frac{1}{2}\nabla^2\varphi_T - \frac{dU_B}{dy} + \frac{1}{2}\frac{dU_T}{dy}). \tag{A.4b}$$

Eq.(A.4a) and Eq.(A.4b) are added and subtracted respectively

$$(\frac{\partial}{\partial t} + U_B\frac{\partial}{\partial x})\nabla^2\varphi_B + \frac{U_T}{4}\frac{\partial}{\partial x}\nabla^2\varphi_T + J(\varphi_B, \nabla^2\varphi_B)$$
$$+\frac{1}{4}J(\varphi_T, \nabla^2\varphi_T) + (\beta - \frac{d^2U_B}{dy^2} - \frac{1}{2}\frac{d^2U_T}{dy^2})\frac{\partial\varphi_B}{\partial x} = -r(\nabla^2\varphi_B - \frac{dU_B}{dy}), \tag{A.5a}$$





$$(\frac{\partial}{\partial t} + U_B \frac{\partial}{\partial x})(\nabla^2 \varphi_T - 2F\varphi_T) + U_T \frac{\partial}{\partial x}(\nabla^2 \varphi_B + 2F\varphi_B) + J(\varphi_T, \nabla^2 \varphi_B)$$

$$+ J(\varphi_B, \nabla^2 \varphi_T - 2F\varphi_T) + (\beta - \frac{d^2 U_B}{dy^2} - \frac{1}{2}\frac{d^2 U_T}{dy^2})\frac{\partial \varphi_T}{\partial x} = -r(\nabla^2 \varphi_T - \frac{dU_T}{dy}), \qquad \text{(A.5a)}$$

In Eqs.(A.5)

$$O(r\nabla^2 \varphi_i) \gg O(r\frac{dU_i}{dy})$$

(Mathew Spydeil and Paola Cessi, 2002; Meng Lu, Lv Ke-li, 2002), where $i = B, T$. Therefore,

Eqs. (A.5) can be reduced to Eqs.(2.3).

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





## LIST OF FIGURES

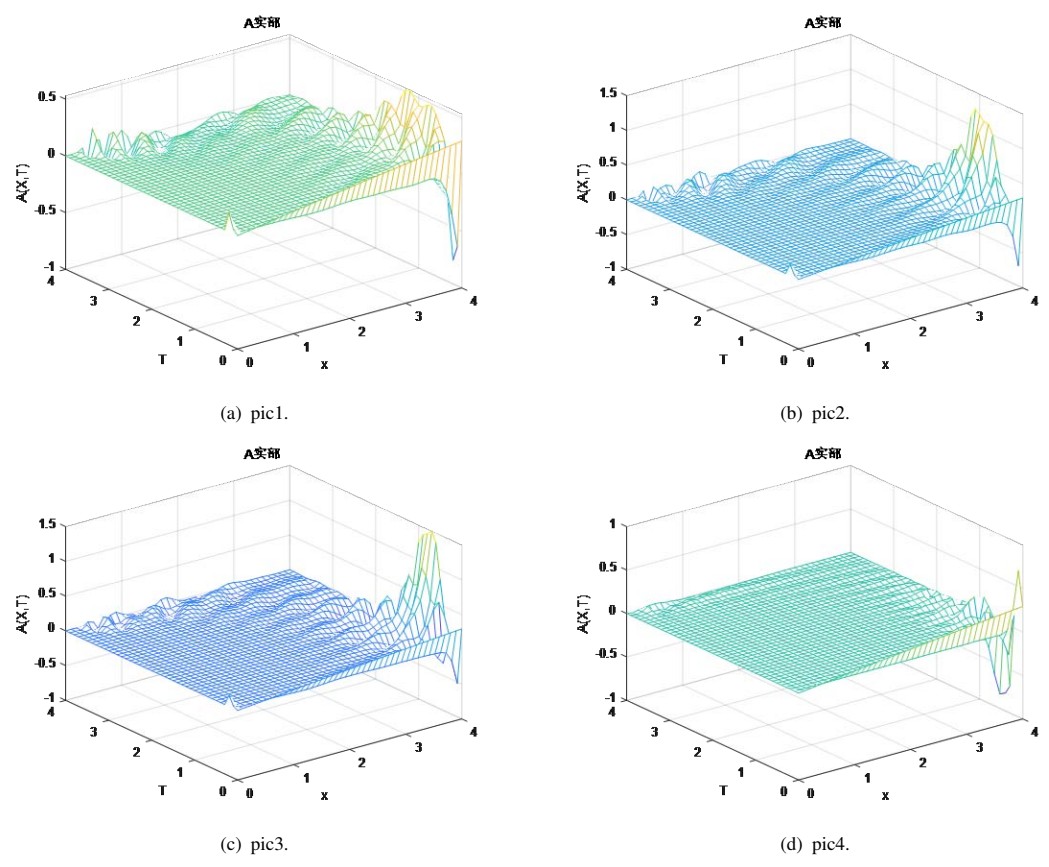

(a) pic1.

(b) pic2.

(c) pic3.

(d) pic4.

FIG. 1. When b is equal to 0.4,.0.8,1.2 and 6,the real part graph of A.


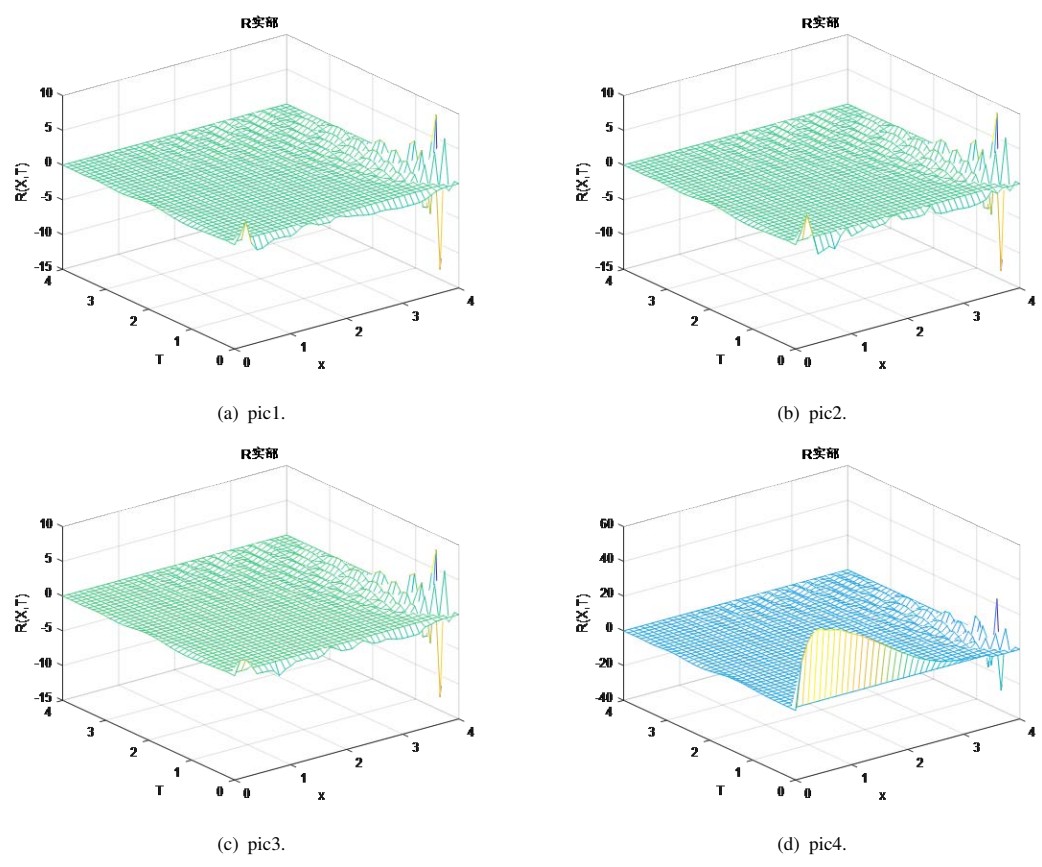

(a) pic1.

(b) pic2.

(c) pic3.

(d) pic4.

FIG. 2. When b is equal to 0.4,.0.8,1.2 and 6,the real part graph of R.