# Peer review of "The Effect of Quadric Shear Zonal Flows and Beta on the Downstream"

_Nonlinear Processes in Geophysics, 2020_

## Referee Comment (RC1) · Anonymous Referee #1 · 26 Nov 2020

I believe that this paper largely duplicates the material in Pedlosky (2019); the only modification here appears to be an introduction of a quadratic zonal shear in the mean flow, which, as far as I understand, boils down to a modified value of the beta-parameter in (2.3) compared to the model without this shear considered originally by Pedlosky. If I am mistaken, the authors should at least clearly delineate how their work is a substantive extension of Pedlosky's ideas. In addition, as of now, the presentation is sloppy and awkward in terms of grammar and style, which will need to be much improved if the paper makes it to publication.

[Figure]

2020-43, 2020.

---

## Referee Comment (RC2) · Anonymous Referee #2 · 3 Dec 2020

The authors study the instability and weakly nonlinear development of a zonal jet having a specific meridional shape. Using the methodology of Pedlosky (2019), they try to generalize his results by including the effects of beta and shear of the background flow on the nonlinear flow behavior. Although weakly nonlinear theory is quite limited in describing the nonlinear development of baroclinic instabilities, such studies can still be useful to discover new physics or dynamics.

However, I suggest to reject the current manuscript because of the following reasons.

1. The manuscript is very immature, in particular in the presentation of the results in section 3 (and figures 1 and 2). It contains also too many typos and errors in language

and formulation.

2. In the context of available recent work (Pedlosky, 2019 and Zhang et al., Tellus A, 72, 1843330, (2020), why is this one not cited?) the result that the weakly nonlinear flow development is influenced by beta and the zonal shear flow (top of section 4), without any more analyses on the physics involved or new dynamics, does not warrant publication.

Some other issues to consider when the authors continue this work:

1. Is the terminology of 'quadric shear' correct? A quadric is an algebraic surface of degree 2. Here the horizontal shear (I think this what the authors have in mind), is linear in y (as $U(y)$ is quadratic in y).

2. When the basic state (2.2) is substituted into the perturbation equations, indeed equations (2.3) result, but these have non-constant coefficients, because they contain terms $U_B$ and $U_T$ which depend on y. Pedlosky (2019) uses constant $U_T$ and $U_B$ in his analysis. The eigenvalue problem for determining the neutral curve in general has to be solved numerically (see e.g. Van der Vaart, Physics of Fluids, 9, 615 (1997)), as traveling wave solutions with $\exp(ily)$ do not exist. Even when considering velocity profiles of the form $U(1 - a y^2)$, with small a, one has to justify that neglecting the $a y^2$ term does not affect the neutral curve.

---

## Author Comment (AC1) · 26 Jan 2021

Dear Editors and Reviewers: Thank you for your letter and for the reviewers' comments concerning our manuscript entitled "The Effect of Quadric Shear Zonal Flows and Beta on the Downstream Development of Unstable Baroclinic Waves" (ID: npg-2020-43). Those comments are all valuable and very helpful for revising and improving our paper, as well as the important guiding significance to our researches. We have studied comments carefully and have made correction which we hope meet with approval. The main corrections in the paper and the responds to the reviewer's comments are as flowing: Responds to the reviewer's comments: Reviewer #1: Response to comment:

[Figure]

In this paper, we mainly refer to the literature of Pedlosky(2019). In the literature of Pedlosky(2019), only the influence of beta effect on the downstream development of barobaric wave is considered, but there are many factors affecting the downstream development of barobaric wave, among which zonal shear flow is one of the factors that need to be considered. After adding zonal shear flow, the influence of zonal shear flow on the downstream development of unstable baroclinic wave is analyzed emphatically. And starting from the final Lorentz equation, by drawing the real parts of A and R respectively, it can be found that the zonal shear flow has a great influence on the baroclinic wave. The writing style is not very mature, there are many aspects need to be improved, we have listened to your advice carefully, efforts to improve my writing level, further improve the language and style of the article. We tried our best to improve the manuscript and made some changes in the manuscript. These changes will not influence the content and framework of the paper. And here we did not list the changes but marked in red in revised paper. We appreciate for Editors/Reviewers' warm work earnestly, and hope that the correction will meet with approval. Once again, thank you very much for your comments and suggestions.

Please also note the supplement to this comment:
https://npg.copernicus.org/preprints/npg-2020-43/npg-2020-43-AC1-supplement.pdf

---

## Author Comment (AC2) · 26 Jan 2021

Dear Editors and Reviewers: Thank you for your letter and for the reviewers' comments concerning our manuscript entitled "The Effect of Quadric Shear Zonal Flows and Beta on the Downstream Development of Unstable Baroclinic Waves" (ID: npg-2020-43). Those comments are all valuable and very helpful for revising and improving our paper, as well as the important guiding significance to our researches. We have studied comments carefully and have made correction which we hope meet with approval. The main corrections in the paper and the responds to the reviewer's comments are as flowing: Responds to the reviewer's comments: Reviewer #2: Response to

comment: Baroclinic instability has always been an issue that scholars have been interested in. By referring to and learning the literature of Pedlosky(2019), it is considered that it is of practical significance to consider the influence of zonal shear flow on baroclinic wave. The research method of Pedlosky(2019) is used for reference and popularized. 1. The name of quadratic shear fundamental flow means that the fundamental flow is a quadratic function of latitude. In "LV Keli , Large orography and barotropic solitary Rossby waves-weak quadric shearing basic flow(in Chinese), Acta Meteorologica Sinica, 1988, 46(4):412-420", the concept of quadric shear was also put forward, that is, the shear of base flow is formed by quadric curve. 2. In the literature of Pedlosky(2019), and are constants. In this paper, It are functions of latitude , respectively. So equation (2.3) has a non-constant coefficient. In this paper, we consider that the basic flow is a quadratic function of latitude. If the form is , with small if is ignored, then becomes a constant, which is the same as the problem discussed in the paper of Pedlosky(2019). 3. We have carefully corrected the spelling errors and language errors in the result part of the third section (including Figure 1 and Figure 2) your mentioned, this part has been modified emphaticallyïijŇand constantly improved the language and style of the paper, so that the paper can reach the best state. 4. We sincerely listen to your opinions and add Pedlosky, 2019 to the references. When submitting this article, the article of Zhang et al had not been published, so it was impossible to quote them. As for the result that the Weakly nonlinear flow development is influenced by beta and the Zonal shear flow (Top of Section 4), we have conducted some further analysis to make the paper more perfect. We appreciate for Editors/Reviewers' warm work earnestly, and hope that the correction will meet with approval. Once again, thank you very much for your comments and suggestions.

Please also note the supplement to this comment:
https://npg.copernicus.org/preprints/npg-2020-43/npg-2020-43-AC2-supplement.pdf

2020-43, 2020.